# The Interdisciplinary Approach to the Conceptualization of the Image of the Arctic and the North in the Mass Consciousness: An Example of Russian Students

Elena Vladimirovna Kornilova 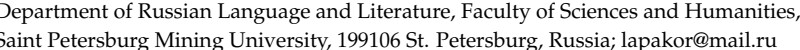

Department of Russian Language and Literature, Faculty of Sciences and Humanities,
Saint Petersburg Mining University, 199106 St. Petersburg, Russia; lapakor@mail.ru

**Abstract:** The article is devoted to the integrated interdisciplinary study of multiple aspects of the conceptosphere of the Arctic and the North and its representation in the collective consciousness of Russian students. The purposes of this paper are to examine the current status of the creation of an integrated paradigm of knowledge about the Arctic in humanities and the social sciences, and to draw conclusions about the emerging trends of understanding of the cultural and symbolic constants of the Arctic area. The respondents (students of Saint Petersburg Mining University) gave 2357 responses to the stimulus "the Arctic" in the course of the free associative experiment. The obtained associates were distributed into culturally significant thematic groups. The main thesis of this study is that the meaningfulness of the image of the Arctic and the North is in many respects determined by key ideas present in the collective consciousness, and by the life priorities and orientations of today's young people. The idea of the extreme, severe conditions, and the danger of this region of the Earth dominates in minds of Russian students. Nevertheless, this land is rich in natural resources and is a unique platform for a wide range of research. A further tendency is identified in the formation of the image of the Arctic and the North in the mass media: new trends are affected by geopolitical factors, focusing on the pivotal role of the Arctic area as a territory of national interests. The creation of a "conflict image" of the Arctic region is counterproductive. In this regard, it is necessary to put into practice the concept of Arctic solidarity, in order to promote integration processes in the exploration of the circumpolar region.

**Keywords:** the Arctic and the North; student youth; image of the site; collective consciousness; conceptosphere; free associative experiment; cultural and symbolic constants



## 1. Introduction

In 2020, Russia approved the *Strategy for Developing the Russian Arctic Zone and Ensuring National Security until 2035*; social and environmental issues are the most important priorities of the new strategy (Strategy 2020). In addition to the challenges and environmental concerns specified in this document of prime importance, a significant place is given to issues of international cooperation, the development of social aspects, and the improvement of living conditions for the population in the Arctic. The year 2021 is marked by Russia's chairmanship of the Arctic Council until 2023. In addition, the Scientific Arctic Council of Russia is being formed under the State Commission for the Development of the Arctic. The Council will coordinate the efforts of various organizations in planning and conducting research projects at high latitudes. The international forums *Arctic: Territory of Dialogue* (Fifth International Arctic Forum, Saint Petersburg, April 2019), and *Arctic: Present and Future* (Tenth Anniversary Online Forum, Saint Petersburg, December 2020) were major platforms for foreign partners' joint discussion on the current problems and the future development of the Arctic. It is worthy of note that the main goal of the Arctic conferences is the discussion of a wide range of issues concerning the sustainable development of

the Arctic and the North from an interdisciplinary standpoint, in respect of new fields of research such as studies of the Arctic and the North.

In the Russian Federation there are over 500 organizations dealing with research in the Arctic: research institutes and branches of the Russian Academy of Science, higher education institutions, companies' research centers, and subordinates of the Ministry for the Development of the Russian Far East and the Arctic. A substantial contribution to exploration of the Arctic and the Antarctic (Litvinenko 2020) has been made by Saint Petersburg Mining University (seventh position in the QS World University Rankings by Subject in 2022 for Engineering—Mineral and Mining), which rightfully is counted among the best-known mining schools in the world (Shchukina et al. 2020): "Saint Petersburg Mining University, the first higher technical educational institution in Russia, will celebrate its 250th anniversary in 2023. This educational institution has played an important role in the history of Russian culture and education" (p. 1012). In 2019, The International Competence Centre for Engineering and Technology of Developing Deposits in Arctic Conditions was launched at the university (Dvoynikov et al. 2020). The author of this current paper is interested in studying this topic within the framework of social sciences, humanities, and linguocultorological studies (Kornilova et al. 2020). This article presents the results of a free associative experiment involving students, and aims to reach conclusions about the emerging trends in understanding of the social aspects of the Arctic area and the conceptualization of the image of the Arctic and the North in the collective consciousness of the younger generation.

Professor Yu. F. Lukin, a leading practicing researcher in the field of the Arctic and Northern European regional studies, is actively promoting the integration of a wide variety of specific studies including problems affecting the development of the Arctic, and the development of an integrated paradigm of knowledge about this unique area of the Earth. Within the disciplines of social science and humanities, his concept (Lukin 2012) of "Arctic solidarity" is of great interest (p. 106), especially in connection with the aggravation of geopolitical problems in the world. Arctic internationalization has been carried out very slowly, as if it had been slowed down by the conventional idea of the utopic development of a transparent, tolerant model of Arctic exploration and the interaction of Arctic countries and other countries in the circumpolar region, acceptable to the global community. However, an approach based on the leading role of the humanities in the understanding of Arctic semiotics can and should be an alternative to the powerful geopolitical instruments recently applied frequently due to the growing interest in the resources, energy, social aspects, economics, and logistic capability of the Arctic region (Cherepovitsyn et al. 2018, 2021).

The notions of the Arctic and the North should be considered concepts of world culture and have been studied within the framework of such complex academic disciplines as Arctic Cultural Studies (by N.M. Terebikhin), and Arctic Regional Studies (by Yu. F. Lukin) and more broadly in the fields of Northern Studies and Arctic Studies. The purpose of these disciplines is to create and to study a coherent image of the Arctic and the North, combining humanities-based components with the natural science aspects of the study and development of this territory.

The *Arctic Thesaurus of Knowledge* is an imposing terminological resource that has emerged in the scientific papers dedicated to the Arctic (Lukin 2019). It provides a set of encyclopedic information and accumulated notions, ideas, and concepts on issues relating to the Arctic within many academic disciplines and sciences. Its objectives are to implement a comprehensive multi-level approach to the study of the current status and evolution of the Arctic and to conceptualize the notions involved. There is no doubt that the Arctic generates the interest of scientists from all over the world, in fields of high technology, geology, ecology, climatology, economics, energy, logistics, etc. (Samylovskaya et al. 2022). Furthermore, the image of the Arctic and the North has also been studied in the contexts of philosophy, history, political science, geopolitics, sociology, psychology, culturology, ethnography, linguistics, study of literature, aesthetics, art history, and other humanities (Kudryavtseva et al. 2020). It is of interest to note that such concepts as "Arctic", "Arctic

region", "Arctic zone", "High North", "Far North", and "Circumpolar North" are used in scientific papers, but globally there is no consistent approach to the definition and correlation of these designations. The most neutral and explicit name is probably the "Arctic region", frequently used in various official documents in Sweden, Finland, and the USA. Within the Russian Federation, the Arctic is understood as part of the North, and the North of Russia as a part of the Arctic, figuratively compared by Lukin to "*matrioshka in matrioshka*" (p. 95) (2012). Leaving aside physical and geographical details of these concepts' definition, such naming units as the Arctic and the North are mainly used in this article within the context of the social sciences and humanities.

In his numerous articles and monographs on the diversity of the Arctic and its representations, Yu. F. Lukin emphasizes that the subject, and more precisely its conceptual notion, should nowadays be enriched with new meanings acquired through interdisciplinary approaches. This helps to improve understanding of the impact of history on the current situation, and if possible to look towards the future (Lukin 2013). The author (Lukin 2019) stated: "The Arctic as an object of research is analyzed and studied by specialists of different sciences and disciplines, because any single branch of knowledge cannot understand, embrace and express conceptually the polyphony of the Arctic. The applying of approaches predominantly in the term of one or another narrow scientific specialization is always similar to a one-side gumboil in the understanding of such a sophisticated object as the Arctic" (p. 5).

Ideally, the image category should have an improved spatiotemporal structure and dynamics. In this sense, the Arctic is one of the "ideal" *images of the site* studied considerably within the frameworks of geography, psychology, psycholinguistics, and many other social sciences. The Arctic is primarily in effect a closed area, representing the physical and geographical region of the Earth adjacent to the North Pole. In addition to specific geographic and landscape characteristics, this region of the Earth has deep historical, cultural, spiritual, and moral roots pertaining to the circumpolar civilization. According to the definition by Yu. F. Lukin, the Arctic of the twenty-first century is neither the form of its existence in the material world with its natural resources, nor the water and the land, but an ethnocultural environment with historical memory, and the value orientations, standards, and traditions of people who inhabit this cold and mysterious land. The author (Lukin 2019) refers to the "spiritual dimension" of the Arctic space, and a special "psychological field" of the Arctic as a part of the psychological field of the global community of the Earth (p. 6).

The works of Terebikhin (Terebikhin and Ovsyannikov 1997; Terebikhin 1998), a famous scholar of philosophy, ethnography, and culture are recognized both in Russia and internationally. His research constructed a holistic concept of the philosophy of culture of the Arctic and the North, including the concept of identifying and interpreting sacred archetypes and symbolic images, which the author defines together as the Metaphysics of the North. He uses figurative means of writing to explain that the North has never been merely a geographical category in the worldview of the Scandinavians and the Russians, and tries to find the way to access the enigma and attractiveness of the Russian North. The *metageography* of the Russian North is associated with diametrically opposed worlds of life and death, existence and other being. According to N.M. Terebikhin, the national consciousness identifies the North and the Arctic as "an insular otherworldly space lying beyond the bounds of existence", and the only way to comprehend this space is to turn away from mundane affairs, material possessions, and short-term interests. He (Terebikhin 2004) imagines the Arctic as a "circumpolar cosmopoietic ring", the world's center where the finite problems of earthly life have been stated and solved (p. 4). There, the world loses its space–time limitation: the light of the polar day makes space thinner and even disappear, while time slows down and becomes eternity (p. 140).

Krasovskaya (2013) studied the issues affecting images built of the North, considering the combination of its geography, rational background and artistic perception: "The image of the North is quite popular in the society and performs objective reality; changes over time and is highly dependent on the worldview of the person or society who creates this

image, its level of culture, ethnicity, etc." (p. 42). Their research drew special attention to the fact that the theory of cultural landscape, established in the early 20th century, has been redeveloped, making possible the combination of features of science and the humanities in this territory, in order to create a clear picture of the Arctic. Likhachev (2004) and Dokuchaev (2012) wrote about the cultural–historical and sacral–symbolic value of the space of the Far North and the Arctic. The Arctic is considered a constantly evolving multidimensional space (Zhuravel 2018), which is characterized by a unique macro-regional identity (Nazukina 2013).

It is important to note that the Far North has always played a significant part in Russian history and culture. It is reflected in a considerable number of "northern" works of fiction by V. Kaverin, K. Paustovskiy, V. Pikul, Yu. German, B. Gorbatov, L. Platov, O. Kuvaev, V. Obruchev, V. Sanin, and other writers. Their works actualize human psychology under the extreme circumstances: endurance and courage, determination and curiosity for exploration of unknown lands and discovery of dangerous routes, desire to be a "pioneer" in various areas of human existence, unbelievable capacity for bravery in the face of hardship and deprivation, and never giving up while maintaining kindness and compassion for others. This issue is comprehensively expressed in the papers of E. Ya. Fesenko, professor of Northern (Arctic) Federal University (city of Arkhangelsk) (Tsvetova 2016). As reported (Fesenko 2016), the idea of a hero who finds himself in difficult conditions is the focus of her research (p. 134). The monograph *The Gravity of the North* (Fesenko 2015) is a unique piece of work from the point of view of social sciences. This work includes an impressive amount of empirical material, with diaries, notes, documents, itineraries, short stories, novellas, and novels, together representing the so-called "northern topic", a metaphysical image of the Arctic space and the North.

In general, an overview of the literature dedicated to the Arctic proves the importance of the integrated interdisciplinary study of multiple aspects of the conceptosphere of the Arctic and the North. Classen (2021) states: "It appears to be a highly exciting new approach in academic research developing bridges between even the most distant fields of investigation in order to reach a higher level of hermeneutics and epistemology, certainly highly desirable for the twenty-first century thirst for new synergies and innovative research" (p. 1). An approach characterized by narrow specialization will neither build the multidimensional model of the Arctic (the so-called metaphysics of the Arctic and the North) nor find cultural and symbolic constants ("the spirit of the place") which are essential for this process of image creation.

## 2. Materials and Methods

It is well-known that a person receives information from many sources to create the image of an object of reality. Predominantly, this involves real-life situations, observations, and interactions with people who can share their knowledge and experience, opinions, and evaluations. However, external sources of information such as books, the mass media, the Internet, cinema, works of art including illustrations and photographs in magazines, as well as educational activities, have played dominant roles in building a generalized image of the Arctic and the North. The entirety of the image comprises pieces of the subjective worldview: phenomena, events, situations, elements of sensual and rational cognition, perceptions, pictures, and associations. Studies of association cognition (Dmitrieva et al. 2020) which is "based not only on the conceptual framework of a language, but on the cultural values" (p. 48) are considered important and very promising. In this respect, the study of semantics and the function of language units in speech based on associative verbal fields has been particularly popular, these fields allowing representation of the process of understanding the world by an individual, and the ways by which some aspects of real life are verbalized in the mass consciousness.

It stands to reason that the process of conceptual image building is protracted and ongoing throughout a person's life since childhood. In this respect, a particularly important period is adolescence during which the personality dynamically develops. It is a period of

transition to independent life, when a person lays the basis of their individual worldview and personal identity. This paper addresses basic features of the associative verbal field of "the Arctic" within the linguistic worldview of young Russian people. The research method used was a free associative experiment: 350 respondents of 18–20 years old were selected at random from students of Saint Petersburg Mining University. This university is well suited for such research. The categories of the selected test subjects are of particular interest, since their prospective careers may be related to the Arctic area (Sharok et al. 2021). The geography of Russia is completely represented at the Mining University; students of different nationalities from many regions study here, representative of more than 200 peoples and ethnic groups living in Russia. It is noteworthy that many students of this university previously lived in the Arctic zone or the Far North. All participants were fully informed about the purpose and anonymity of the research, as well as about the use of the data obtained and the absence of any risks associated (ethical approval is not required for this type of study).

In each case, the experiment was run for 5 min; the number of responses within 5 min was not limited. The aim of the experiment was to reveal the diversity of the associative connections in response to the stimulus "the Arctic", and to identify the prevailing semantic components of perception of this concept in the respondents' collective consciousness, taking into consideration the frequency of the responses, which were classified into thematic groups. During the experiment the respondents were asked to present individual written responses to the stated word "the Arctic", namely, the first words that came to mind, collocations, or phrases, including proper names, toponyms, naming terms, and precedent texts. The respondents did not face any obstacles, nor did any refuse to complete the word association.

Free associative experiment is a tool for well respected, simple, and easy-to-perform analysis for use in psycholinguistics, cognitive linguistics, and cultural studies, which are the most promising and dynamic branches of humanities and social sciences. The method was first attempted in 1879 by Sir Francis Galton, the founder of differential psychology and psychometry (Galton 1879), and the test offers an excellent opportunity to study the sophisticated process of the world cognition within an individual, and to represent the essential aspects of the conceptosphere of one's inner world. Pishchalnikova (2019) notes: "Associative experiments uncover people's active attitude to the world represented by language means that determines their relevant strategies of verbal activity and mediates the specifics of their world conceptualization" (p. 749). Such studies (Sedykh et al. 2019) have traditionally been of strong interest in the context of the problem of identifying the dominant features of national linguistic identity (p. 9039).

The research hypothesis is as follows. *The Arctic and the North* refers to basic concepts with extremely broad semantics; it is the most important conceptual area in the mass consciousness. The structure of this field encompasses not only ethnocultural but also universal elements. Linguocognitive analysis of verbal associates representing this conceptual field makes it possible to identify and describe stereotypical ideas about the Arctic region that exist in the worldviews of various social groups. This research focuses on the free associative verbal network of the responses (by student youth) to this image symbol, allowing further analysis of the stereotypical representation of this concept and prompting certain conclusions about trends in its conceptualization.

### 3. Results

The respondents gave 2357 responses to the stimulus "the Arctic" in the course of the associative verbal experiment. The very first, spontaneous responses are the most significant. This paper focuses on the comparison of the responses in order to reveal the most frequently stated; those most common among all the students are known as the "core" associates. All the received associates were divided into relevant thematic groups, in descending order of total percentage of responses (The data are shown in Figures 1 and 2):

(1)　psycho-emotional responses (38.7%);

(2)  responses referring to research (27%);
(3)  responses representing Arctic toponyms and other proper names (17.6%);
(4)  responses representing Arctic biota and the Arctic community (15.7%);
(5)  responses with a geopolitical component (1%).

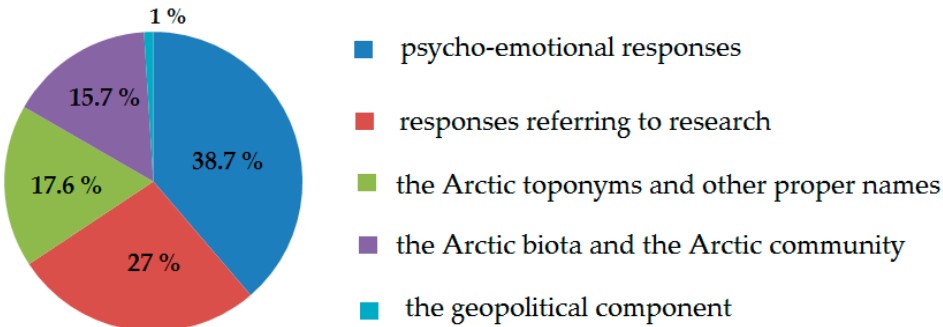

**Figure 1.** Thematic groups for the associative verbal field "the Arctic".

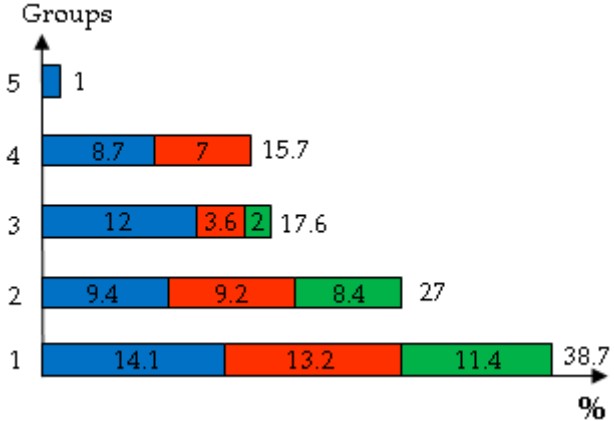

**Figure 2.** Percentages of thematic groups.

The list below states the most common responses of the respondents, and single associations. In calculating the percentage of thematic groups, the number of repetitions among the common responses was taken into account (individual associations were stated in almost all the responses).

1. Psycho-emotional responses—feelings, senses, and assessments as well as stereotypical ideas associated with the conceptual field "the Arctic" (total 38.7%):

–  perception of the Arctic as a cold, snow-covered area with extreme climatic conditions—14.1%: *cold, frost, snow, cold land, snow kingdom, snow/Arctic desert, avalanches, snowfall, snow storm, blizzard, wind, snowdrifts, ice crust, winter, ice, centuries-old ice, blocks of ice, icebergs, glaciers, ice floes, endless frosts, shivering with cold, hypothermia, frostbite, low temperature, rime on the eyelashes, severe weather conditions, severe/unsuitable for life conditions, extreme;*

–  common names and emotional and sensual definitions of the Arctic area, and words describing how a man feels when he is there—13.2%: *endless space, high latitudes, lifeless land, uninhabited islands territory, field, emptiness, empty, nothing, a zone of the Earth, the end of the world, distance, faraway, remoteness, inaccessibility, hard to reach, eternity, adaptation, struggle, hardening, survival, life, death, destruction, life at the limit, bewitching, beautiful, beauty, beautiful winter, voidness, despair, alienation, calmness, detachment from the world, solitude, loneliness, isolation, danger, hunger, illness, difficulties, unusual, silence, stillness, calm;*

–  elements of color, light, and time perception—11.4%: *whiteness, white, snow-white color, white desert, white snow, white captivity, white beard, red nose, red ski overalls, aurora,*

*dazzling, bright sun, blue, dark blue, blue/transparent/pure ice, clear sky, dark, darkness, obscurity, light, gas lantern light, polar night, eternal night, long night, polar day*.

2. Responses referring to research (total 27%):

–    associations including research semantics and difficulties people faced while exploring the Arctic, as well as branches of science—9.4%: *science, scientific/challenging research, researchers, scientists, expedition, Soviet expeditions, polar explorers, navigators, discoverers, pioneers, rescuers, fanatics, shift, work, hard work, Arctic exploration, geographical/scientific discoveries, North Pole discovery, magnetic pole, records, long business trips, adventurers, mystery, mysterious, unknown, little-known, dream, study, tests, interest, history in icepacks, films, documentaries, museum, travel, excursions, geology, geography, glaciology, climatology, meteorology, ecology*;

–    natural resources of the Arctic, specific areas, objects and methods of research, technical devices, floating structures, aircraft, and related associations—9.2%: *shelf, shelf zone, subsoil, natural resources, minerals, mineral deposits, oil, gas, liquefied gas, gold, wells, well drilling, gas production, exploitation, mountains, water, fresh water supply, water rise, sea level, currents, glaciers melting, ice/icebergs movement, climate, climate change, global warming, stars, bacteria, microorganisms, suspended animation, study of anthrax, nuclear tests, nuclear test site, new trade routes, shipbuilding, icebreakers, nuclear-powered ships, ships, cruisers, submarines, airplanes, helicopters, transarctic flights, air crashes, sea tragedies*;

–    research stations, the realities of life and daily routine of polar explorers, details of clothing, equipment, etc.—8.4%: *scientific center, station, polar stations, weather station, settlements, tents, equipment, Eskimo fur boots, valenki, mittens, warm gloves, fur, deer skins, fur/warm clothing, woolen sweater, warm boots, overalls, down jacket, hood, fur coat, hat, hat with earflaps, scarf, thermal underwear, snowshoes, snowmobiles, ski, sledges, dogs, laika, husky, compass, globe, map, gas lamp, lantern, thermos, hot tea*.

3. Responses representing Arctic place names and proper names associated with the Arctic (total 17.6%):

–    names of geographical or astronomical objects and administrative units of the Arctic region, including the diachronic aspect along with interconnecting and contrasting names—12%: *Arctic Ocean, North Pole, North, Northern Hemisphere, Northern Sea Route, Arctic Circle, Spitsbergen, Greenland, Novaya Zemlya, Kola Peninsula, Barents Sea, White Sea, Kara Sea, Bering Strait, Chukotka, Alaska, Yakutia, Yamal, Siberia, Khibiny, Russia, Soviet Union, Finland, Canada, Norway, Scandinavia, Murmansk, Arkhangelsk, Apatity, Surgut, Antarctica, Earth, Polar Star, constellation Ursa Major, Milky Way*;

–    proper names and precedent phenomena indirectly associated with the Arctic and the North—3.6%: *the Russian Geographical Society, the icebreaker "Krasin", the nuclear-powered icebreaker "Lenin", the shipwreck of "Chelyuskin", "Titanic", "Mishka in the North" [chocolate], Umka [little white bear, a character from a Soviet cartoon], the song of 'Mamontionok', the Baby Mammoth [a cartoon character traveling on an ice floe in search of his mother], "Two Captains" [V. Kaverin's novel], "Lost in the Ice" [Icelandic film of 2018], "Territory" [Russian film of 2014 based on the novel by O. Kuvaev], "The Chukchi is Waiting for Dawn in his Chum" [lyrics from a song], "I'll Take You to the Tundra" [popular Soviet song]*;

–    names of travelers, sailors, explorers of the Arctic, certain Arctic lands, and their derivatives—2%: *Semyon Dezhnev, Ferdinand Wrangel, Fedor Matyushkin, Otto Schmidt, Robert Peary, Vitus Bering, James Cook, Alexander Fersman, and members of the crew of "Chelyuskin" known as Chelyuskintsy*.

4. Responses representing the Arctic biota and the Arctic community (total 15.7%):

–    representations of the nature, flora and fauna of the Arctic—8.7%: *permafrost, wildlife, desert, no soil, ocean, sea, tundra, taiga, no plants/trees, poor vegetation, no forest, no flowers, endemic plants, polar bears, reindeer, Arctic fox, polar wolf, predators, foxes, amazing animals, sea inhabitants, elephant seal, seals, walruses, fish, whales, killer whales, penguins/no penguins, gulls, goose*;

–  representations of people in the Arctic, their relationships, indigenous peoples and their traditional dwellings, and ethnocultural realia—7%: *people, narrow-eyed people, few people, deserted, a man with a frozen beard, solidarity, friendship, northern peoples, northerners, a Siberian man, reindeer herders, Yakuts, Eskimos, Khanty, Chukchi, tents, yurt, igloo, chum, hunting, dog sled, sleigh, sledges, boats, raw/dried fish, venison, stroganina— sliced frozen fish or venison.*

5.  Geopolitical components were included in only 1% of responses: *international territory, a sector of the territory, politics, cold war, military confrontation, domination, international life, cooperation.*

## 4. Discussion

The associations (2357 responses) effectively represent the associative verbal field "the Arctic" in the collective consciousness of the young people (Saint Petersburg Mining University students) (see Figure 3). Analysis of the results of the free associative experiment showed that the *core zone* of the associative verbal field comprised the first and second thematic group, including psycho-emotional responses and semantics of scientific research. The *near core zone* was composed of the third and fourth groups, representing proper names (particularly Arctic place names), precedent phenomena, and references to Arctic nature and students' vision of the people in the Arctic. The geopolitical component was represented by a negligible number of single responses which formed the *periphery* of all the associations.

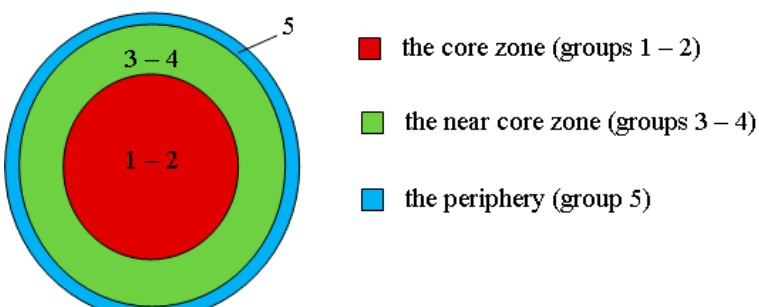

**Figure 3.** The associative verbal field "the Arctic" in the collective consciousness of student youth.

In general, the insignificant number of individual sporadic responses reveals the low individualization of visions of the Arctic in the collective consciousness of the respondents. The most frequent associations were: *cold, frost, snow, ice, glacier, icebreakers, permafrost, winter, water, sea, North, North Pole, Arctic Ocean, Arctic Circle, polar night, northern lights, icebergs, white (color), polar bears, penguins/no penguins, polar explorers, expeditions, melting glaciers, natural resources, minerals, scientific research/discoveries, polar stations, unknown/unexplored, few people, danger, snowstorm, blizzard (and various types of snowfall with wind), warm clothing (and specified items of clothing: fur coat, hat, overalls, etc.)*. However, the core associations stated in the majority of responses were lexical units describing the severe Arctic climate. Alongside these were associations correlating with the image of the Arctic as a source of environmental assets, research potential, and exploitation of mineral resources.

As the core zone of the associative verbal field comprised two thematic groups, the following conclusion can be made: the Arctic as one of the dominant elements of the collective consciousness and a homonymic basic concept is, in the minds of students, a synonym of the cold, eternal ice, snow, and mineral reserves. The image of the Arctic is associated with white color (*white desert, white snow, white captivity*, etc.), and with a certain emptiness and remoteness, reflecting the inaccessibility of the place. Taking into account all the most frequent associations stated in the thematic groups, the overall picture of a stereotypical vision of the Arctic region involves polar bears, reindeer, polar wolves, Arctic foxes, and other predators roaming the endless snow-covered expanses of the Arctic, while seals, walruses, and whales live in the seas. Apparently, according to some respondents,

there are penguins in the Arctic (indicating an associative connection between the Arctic and the Antarctic), although more stated "no penguins". There are few people in this region, but the area is not completely uninhabited: the Chukchi, Eskimos, Yakuts, Khanty, and other indigenous peoples live in yurts, chums, and snow dwellings (igloos), they mainly engage in hunting and fishing, ride dog sleds, and eat sliced frozen fish or venison. People who live and work here need warm clothing. In addition, vegetation is sparse in the Arctic and there are almost no trees and no soil; here the polar nights are long, and you can see a unique natural phenomenon, the aurora.

The vision of the Arctic as a physical and geographical region of the Earth has been formulated quite clearly in the collective consciousness of modern youth. Many of the coincidental associations are geographical markers, i.e., *The North Pole, Arctic Ocean, Arctic Circle*, or toponyms such as the names of islands, peninsulas, archipelagos (*Spitsbergen, Greenland, Novaya Zemlya, Yamal, Kola Peninsula*), seas or straits (*Barents Sea, White Sea, Kara Sea, Bering Strait*), mountains (*Khibiny*), large areas and regions (*Siberia, Alaska, Chukotka, Yakutia*), or northern Russian cities (*Murmansk, Arkhangelsk, Apatity, Surgut*). A considerable number of responses reflected the etymology of the word "*Arctic*"—"a place located under the constellation Ursa Major" which can be used as a navigational pointer towards the Pole Star. The responses included branches of the sciences (*geology, geography, glaciology, climatology, meteorology, ecology*) and some terminological names (*anabiosis, endemic plants, shelf, liquified gas, etc.*) that can be explained by the general profile of the respondents who were students of the Mining University. In the students' collective consciousness, the image of the Arctic is inseparable from notions of research, which became evident due to the frequent use of the associations *scientific research/discoveries, expeditions, polar stations*, and also the mention of specific research objects, devices, etc. The age of the respondents led to the choice of such precedent names as the cartoon characters *Umka* and *Mamontionok* (*Song of Mamontionok*). It is interesting that the nomination *Titanic* was also quite frequent, apparently explained by its associative connection with the word "*iceberg*", the cause of the shipwreck.

It is important to note that the axiological vectors of the responses were positive and negative assessments associated with the Arctic. Among these, the diametrically opposite associates are "*life*" vs. "*death*", and "*survival*" or "*life at the human limits*" vs. "*death*". In general, negatively defined associations prevailed, representing the general assessment of human activity in the Arctic—"*estrangement*", "*danger*" and "*difficulties*".

## 5. Conclusions and Study Potential

In conclusion, it can be noted that for a fully-fledged interpretation of the mental image of a place, i.e. "the spirit of a place", in particular the Arctic space, it seems extremely important to identify not only specific spatial coordinates, event lines, personalities, and artistic symbolism (literature, painting, music, cinema, photography, etc.), but also a communicative and linguistic component—symbolic words, stable expressions, precedent texts, and verbal associations that allow representation of significant fragments of reality in the mass and group linguistic consciousness. In works on discourse analysis (Wood and Kroger 2000), the special nature of language and its close connection with the key problems of the social sciences have been noted. In this regard, the verbal association method is very informative.

The summary of the results of the free associative experiment has shown that the idea of extreme, severe conditions, and the remoteness and danger of this region of Earth dominates the visualization the Arctic in the minds of student youth. Nevertheless, this land is rich in natural resources, offering a source of minerals and a unique platform for a wide range of research. The main semantic and cognitive features of the conceptual field "the Arctic" are "cold" and "extreme nature and climate conditions, unhospitable and unsuitable for human life and economic activities". It is interesting that the test results were consistent with the generalized characteristic of the area described by Professor

Lukin (2012): "the Arctic is a perfect example of the domination of *Thalassocracy*, i.e., the domination of water, ice, sea and cold clean air" (p. 96).

One important conclusion refers to the shift of the ideological connotations relating to Arctic exploration. In the Soviet times, the concept of the Arctic was associated with victory, heroism, patriotism, and great achievements of the country (Zhigunov 2019). Certainly, canonic discourse about acts of heroism by polar explorers had compelling reason. The history of Arctic exploration by the Soviet Union in the early part of the 20th century was heroic. Precisely due to experiences in the Arctic, the highest state award of the title of Hero of the Soviet Union was established in 1934. The first people decorated were "seven braves", pilots who rescued 107 passengers and crew members of the "Chelyuskin" crushed by ice in the Bering strait (Turkov 2017). An American researcher Petrone (2000) referred to the press coverage of the "Cheluskin epic" as a particular communication technology. However, the names of the polar heroes and the Arctic explorers (*I. Papanin, V. Chkalov, S. Obruchev, N. Urvantsev,* etc.) are unfamiliar to young people, as was evident from their responses. There was no nominated association "*hero*" (although "*pioneers*", "*pathfinder*", "*rescuers*", and "*fanatics*" occurred) and the *Cheluskintsi* were mentioned only once, as were the wreck of the "*Chelyuskin*" and the names *O. Smidt, S. Dezhnev, F. Vrangel,* and *F. Matushkin.*

Thus, in this research, the method of verbal associations was supported by semantic–cognitive analysis and evaluation of the characteristics of units obtained during the experiment. In order to improve the verification of the experimental results, there are plans to involve other universities in the research in future, and to conduct a sociological survey among students of different specialties.

Certainly, the overall picture of the representation of the image of the Arctic and the North in the collective consciousness of young people and in the mass consciousness should be studied by various methods, employing the full range of tools used in psycholinguistics, cognitive linguistics, culturology, philosophy, sociology and other disciplines of humanities and the social sciences. The methods and tools applied should be improved. In this respect, any experience of such experiments can be interesting. For example, in the region of Valencia (Spain) a survey was conducted of primary and secondary school children as well as older students to identify the main information channels through which they receive information on climate change (Morote and Hernández 2022).

Thus, when studying the psychological aspects of the building of spatial images at one of the schools in Archangelsk city, Solovyova (2010) conducted a survey using elements of the word association test to identify the image of the Arctic in the minds of adolescents aged 14 to 15 and identified the sources of mental images of the region: fiction, scientific literature, fine arts, cinema, mass media, etc. Analysis of the survey results allowed the author an opportunity to discover that "the image of the Arctic amid teenagers is undifferentiated, generalized, poorly detailed, indistinct and one-sided", and "the image volume is not sufficient" (p. 95).

In the future, it is advisable to expand the practice of such studies into different social and age groups. In general, scholars involved in humanities and the social sciences have demonstrated an interest in the problem of conceptualizing the image of the Arctic and the North, while the studying the visualization of these conceptual fields in the individual, group, and mass consciousness. This academic interest is currently acquiring a particular relevance in the context of the preservation of cultural and humanistic values (Stepanova and Shchukina 2022). The image of the Arctic and the North has been modified in the course of the region's economic development, and the polar expeditions providing relevant databased information have contributed significantly to the image change. This image has developed from the artistic and mythological picture of the "sacred" virgin land, remote from civilization and ideal for spiritual meditation, towards the rational, practical and even pragmatic image of an area open to explore, conquer, and harness by humans who are explorers and sovereigns of nature. To provide a better understanding of the peculiarities of the Arctic circumpolar civilization's development, it is advisable to use *indigenous methodology* (Kovach 2021), which aims to enrich science with the worldview and

experience of indigenous peoples, including a deep vision of the prospects and interests of these peoples provided by their own scientists (indigenous scholars).

The characterization of the Arctic as a geopolitical space is also of great interest to modern science. Interestingly, according to the results of the associative experiment, the analysis of this component is difficult, since it was represented by a negligible number of single responses (1%). Apparently, this is due to the young age of the respondents (18–20 years old), whose interests are mainly focused on educational activities. As Crowther-Heyck (2005) noted: "If a concept could be operationalized, then it did have a place in science" (p. 65). However, from the standpoint of operationalization in the social sciences, the measurement of a fuzzy concept is possible with the help of empirical observation, especially if its existence is implied by other phenomena. In this case, not only can quantitative indicators act as units of measurement, but certain assessments, opinions, and actions are also useful. In this context, it is important to note the following. Currently, Arctic megatrends are given an increasingly significant place in print media, online publications, and the news content of various news agencies. It seems obvious that the image of the Arctic and the North as a resource storehouse and a territory of geopolitical disagreement has been dynamically developed in the public consciousness especially through the mass media (Kovrigina 2015).

There is a tendency in contemporary media space to give prominence to geopolitical factors, focusing attention on the strategic importance of the Arctic region as a zone of national interests. In the media space, in addition to "*research*", "*prospects*", "*monitoring*", "*optimization*", "*development*" and "*exploration*", there are other basic semantic constants for the Arctic, such as "*challenges*", "*menace*", "*disputes*", "*conflicts*", "*danger*", "*hazards*", "*rivalry*", "*controversy*", "*military involvement*" and a "*struggle for resources*". In many papers that contribute to the media image of the Arctic, strong expressions such as "*battle for the Arctic*" and "*Arctic race*" have been used (Scherbinin and Danilova 2014; Konyshev et al. 2016).

Thus, the representation of the Arctic and the North in the mass media can be considered as distinct direction of research. It seems that the creation of a "conflict image" for the Arctic region is counterproductive, since it cannot contribute in any way to the consolidation of the global community's efforts to solve through cooperation the problems that humanity faces. As Lukin (2012) reasonably noted: "*The Arctic, Universe, the World Ocean*, the implementation of these and other global projects today is impossible without the global human solidarity, without the use of the intellect of mankind, without the cooperation of all types of available resources, including the development of the advanced infrastructure in the Arctic. Therefore, perhaps the most important conclusion is that it will be difficult to explore the Arctic without a dialogue between people and cultures" (p. 109).

It is of great importance to further study the principles of constructing the image of a real fragment of reality in the collective consciousness of the younger generation and in the public mind in general. The interdisciplinary approach to conceptualizing of the image of the Arctic and the North, strengthening the contributions of social sciences and humanities in understanding the semiotics of the Arctic, will contribute to the development of integration processes during territorial development [an example of a positive experience is the Russian–Norwegian cooperation (Smirnova et al. 2018)]. In the context of global trends in training personnel for the Arctic, qualitative changes are needed in the Russian educational system. In Russia's leading industrial universities, the formation of basic knowledge in natural sciences should be accompanied by the study of the socio-cultural specifics of the Arctic region (Samylovskaya et al. 2020). Simultaneously, the so-called Arctic imperative should be the key driver of the special attitudes of the entire global society to the problem of preserving the environment, human capital, and ethno-cultural values of the circumpolar region. The Arctic imperative provides for the harmonization of man with the environment in the geoclimatic conditions of the Arctic, within the triad "*humanity–society–biosphere*". Recognizing this factor, the scientific community will contribute to a comprehensive study of the Arctic circumpolar civilization as a historical phenomenon in the development of mankind.

**Funding:** This research received no external funding.

**Institutional Review Board Statement:** Not applicable.

**Informed Consent Statement:** Not applicable.

**Data Availability Statement:** Not applicable.

**Conflicts of Interest:** The author declares no conflict of interest.

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
