# Peer review of "The Interdisciplinary Approach to the Conceptualization of the Image of the Arctic and the North in the Mass Consciousness: An Example of Russian Students"

_socsci, doi:10.3390/socsci11120580_

Round 1

Reviewer 1 Report

It is recommended to state the research hypothesis more clearly.

Taking the operationalization model as a benchmark, how was the interdependence between collective consciousness and policies established? An objective clarification would be welcome.

A more diversified bibliography from the area of the Arctic Circle nations would have qualitatively ensured the final discussion when you refer to the need for cooperation and the existence of multiple multicultural spaces.

Author Response

Please see the application.

Reviewer 2 Report

The Problem of the Interdisciplinary Approach to the Conceptualization of the Image of the Arctic and the North in the Mass Consciousness Overall, this is a solid paper - well-researched, well-presented, built around a survey of free-associations by Russian students in order to exhibit how the arctic is imagined in (Russian) popular culture or „mass consciousness“. The results of this study are not surprising, their presentation is nevertheless interesting to read. — However, the underlying argument or thrust of the paper needs to be clarified. The paper begins with an impressive literature review that highlights the emergence of an interdisciplinary understanding of the arctic. The paper‘s title suggests that there might be a problem with it (but perhaps it refers to the problem that motivates and justifies this interdisciplinary approach). There is no discussion of „the problem,“ however. Instead the investigation of popular consciousness is juxtaposed to the theoretical opening of the paper - there is only an implicit suggestion that popular understanding is quite stereotyped and that it would benefit from the interdisciplinary comprehensive research-effort described at the beginning. It remains a bit of a riddle how these elements of the paper are related and what the authors call for. At the very end there is a reference to the „arctic imperative“ which also needs to be motivated and explained. ((As for English - the paper is written in competent readable English but needs to be checked by a native speaker especially regarding use of the definite article.))

Reviewer 3 Report

The manuscript is an interesting case study on Arctic perceptions based on a survey among Russian students. It is academically and pedagogically interesting, and original as almost all literature is by Russian scholars. Though the number of responses, ie. sampling, is large, they represent only one university in Russia. As outcomes, most of the responses are not surprising (1st & 2nd thematic groups), some ones even obvious (synonyms of cold, ice, snow, white). Thus, the contribution to scholarship is not that high, in particular as foreign literature on (Arctic) perceptions, images, identities is almost neglected, and neither discussed nor reflected. However, it is interesting that, as only 1% has geopolitical component "a 'conflict image' of the Arctic region" can interpreted to be "counterproductive", as the authors conclude. I would recommend highlighting this conclusion even more, and also mention it in the abstract. Also the word "problem" in the title is misleading, the word "issue" would fit better. Following from this, I would this contribution could be interesting if there will some reflections to international discussion on Arctic perceptions & images, and references of foreign literature. 

Author Response

Please see the application.
